# Associations between Natural Resource Extraction and Incidence of Acute and Chronic Health Conditions: Evidence from Tanzania

**DOI:** 10.3390/ijerph18116052

**Published:** 2021-06-04

**Authors:** Isaac Lyatuu, Georg Loss, Andrea Farnham, Goodluck W. Lyatuu, Günther Fink, Mirko S. Winkler

**Affiliations:** 1Environmental Health & Ecological Sciences, Ifakara Health Institute, Dar es Salaam P.O. Box 78373, Tanzania; 2Department of Epidemiology and Public Health, Swiss Tropical and Public Health Institute, P.O. Box, CH-4002 Basel, Switzerland; georg.loss@swisstph.ch (G.L.); andrea.farnham@swisstph.ch (A.F.); guenther.fink@swisstph.ch (G.F.); mirko.winkler@swisstph.ch (M.S.W.); 3University of Basel, P.O. Box, CH-4003 Basel, Switzerland; 4Programs Department, Management and Development for Health (MDH), Dar es Salaam P.O. Box 79810, Tanzania; gbahaty@gmail.com

**Keywords:** DHIS2, health facility disease diagnoses, health impact assessment, natural resources extraction, routine health management information system

## Abstract

Natural resource extraction projects are often accompanied by complex environmental and social-ecological changes. In this paper, we evaluated the association between commodity extraction and the incidence of diseases. We retrieved council (district)-level outpatient data from all public and private health facilities from the District Health Information System (DHIS2). We combined this information with population data from the 2012 national population census and a geocoded list of resource extraction projects from the Geological Survey of Tanzania (GST). We used Poisson regression with random effects and cluster-robust standard errors to estimate the district-level associations between the presence of three types of commodity extraction (metals, gemstone, and construction materials) and the total number of patients in each disease category in each year. Metal extraction was associated with reduced incidence of several diseases, including chronic diseases (IRR = 0.61, CI: 0.47–0.80), mental health disorders (IRR = 0.66, CI: 0.47–0.92), and undernutrition (IRR = 0.69, CI: 0.55–0.88). Extraction of construction materials was associated with an increased incidence of chronic diseases (IRR = 1.47, CI: 1.15–1.87). This study found that the presence of natural resources commodity extraction is significantly associated with changes in disease-specific patient volumes reported in Tanzania’s DHIS2. These associations differed substantially between commodities, with the most protective effects shown from metal extraction.

## 1. Introduction

Implementation of natural resource extraction projects often triggers a series of complex environmental and social–ecological changes [1,2,3,4]. These changes include increased population growth and urbanization, infrastructure improvements, movement and installation of heavy machinery, changes in land use, increased business and economic opportunities, and household resettlement [5,6,7]. Such changes can positively or negatively affect the health status of communities in proximity of resource extraction activities [8,9,10,11,12]. Studies have linked activities of the resource extraction with increased incidence of respiratory diseases [13,14]; sexually transmitted infections, including HIV/AIDS [15]; malnutrition [16]; vector-related diseases [17,18,19]; mental health [20,21]; and cancer diseases [9]. On the other hand, resource extraction projects can positively contribute to population health through improved labor market opportunities and corporate social responsibility (CSR) activities that support local health systems or contribute to general health and education programs [22,23,24,25].

Even though a growing literature has highlighted the importance of comprehensive health impact assessments (HIA) of mining projects [26,27,28], evidence on the impact of large-scale mining projects on population health remains limited. One of the key challenges for such projects is the lack of reliable data sources for local population health [4,29,30]. Due to major international efforts, such population-level health data are increasingly becoming available at the health facility level in low- and middle-income countries, often supported by new digital systems. DHIS2 (District Health Information Systems) is one such web-based software application that can be integrated into the national health management information system (HMIS) to facilitate data collection, data use, data management, and archiving of routine data [31]. DHIS2 has been installed in over 73 countries worldwide [32], allowing researchers and policy makers to collect and aggregate data across the health system [33,34,35].

In this study, we used the data from Tanzania’s DHIS2 to evaluate the association between the presence of different types of natural resource extraction projects and disease incidence at the district level.

## 2. Materials and Methods

### 2.1. Study Setting

Tanzania has an estimated total population of 57 million people [36] and ranks 163 out of 189 countries and territories with a human development index value of 0.529 [37]. Tanzania has a decentralized health system with administrative units in the mainland area extending to 26 regions, 184 districts and municipal councils, and 8941 health facilities (including both private and public health facilities). According to the Institute of Health Metrics’ (IHME) Global Burden of Diseases estimates, the top three Disability Adjusted Life Years (DALYs) per 100,000 population in Tanzania between 2015 and 2019 were consistently shown as (1) neonatal disorders, (2) lower respiratory infections, and (3) HIV/AIDS [38].

Tanzania has a long history of resource extraction activities [39] and was one of the earliest DHIS2 implementers after the platform was developed in 2007 [40,41].

### 2.2. Study Design

This is a multi-year cross-sectional study that aims to assess the associations between the presence of district-level commodity extraction and disease diagnoses reported in Tanzania’s DHIS2.

### 2.3. Data

Disease diagnoses data come from the DHIS2 outpatient department (OPD) dataset indicators and one additional indicator (i.e., total clients tested and found positive for HIV) from the HIV testing and counselling dataset. As part of national HMIS, disease diagnoses made during health facility visits are captured and summarized in HMIS books, and summary information is entered into the DHIS2. We extracted annual aggregated data at the district level covering the period from 2015 to 2019. The data contain counts of disease diagnoses from health facility visits (both public and private). We excluded laboratories, maternity homes, and all facilities that do not provide general OPD services. We excluded the region of Dar-es-Salaam (with 5 administrative districts) due to the high density of population and health facilities and the absence of large-scale resource extraction projects. We merged data from the Ifakara town district and Kilombero District to accommodate available shapefiles in the geographical information system (GIS), whereby Ifakara town District was formally part of Kilombero District until 2016. Our final dataset contained a total of 178 districts.

Data on natural resource extraction activities were accessed via the mineral occurrence map of Tanzania [42], which was based on the Geological Survey of Tanzania (GST) conducted in 2015 [43]. The GST lists a total of 480 resource extraction projects with location coordinates and commodity types. In this list, 42% of the projects are marked prospective, 20% active, and 38% inactive. We restricted our analysis to natural resource extraction projects that were classified as active in the GST as of 2015 (at the beginning of our sample period).

### 2.4. Variables

#### 2.4.1. Exposure Variables

The primary exposure variable of interest was the presence of a given commodity extraction in the district. We considered three types of commodity extraction: (1) construction materials, (2) gemstone, and (3) metals. As shown in Figure 1, most districts contain only one type of extraction project; multiple types of extraction projects were found in only seven districts. Given that coal was identified in only one district and hydrocarbons are mostly located offshore, these two groups were excluded from our analysis. A complete list of the specific commodities extracted is available in Appendix A (Table A1).

#### 2.4.2. Outcome variables

We selected and grouped disease indicators based on the environmental health areas (EHA) framework defined in the HIA guidelines developed by the International Finance Corporation (IFC) [44]. The EHA framework provides a conceptual linkage between resource extraction activities and potential community-level impacts, incorporating a variety of biomedical and key social determinants of health [45]. For this study, we divided all reported health programs in the DHIS2 into disease groups based on the EHA framework. These groups, as well as the OPD indicators falling into each group, are summarized in Table 1.

### 2.5. Statistical Analysis

To assess the association between the presence of commodity extraction projects and disease incidence, we used Poisson regression models controlling for period (in years), population, region fixed effects, and number of health facilities by time in the district. We assumed all commodity extraction projects to remain active over the full sample period. We employed cluster-robust standard errors to correct for residual correlation at the district level over time as well as over dispersion. We report incidence rate ratios (IRR) and corresponding 95% confidence intervals (CI) from the final models. All analyses were implemented using the STATA 15.0 statistical software package [46].

### 2.6. Ethical Considerations

This study obtained ethical approval from Ifakara Health Institute Review Board and the National Institute for Medical Research (NIMR) in Tanzania; the Ethics Committee of Northwestern and Central Switzerland (Ethikkommission Nordwest- und Zentralschweiz, EKNZ); and the institutional review board of the Swiss Tropical and Public Health Institute (Swiss TPH) in Switzerland. 

## 3. Results

### Participants: Overall Characteristics

A summary overview of the number and type of health facilities and commodity extraction projects in districts in Tanzania is shown in Table 2. On average, a district contained 1 hospital, 5 health centers, 36 dispensaries, and 1 health clinic. A total of 2 districts were exposed to all 3 types of commodity extraction, 5 districts were exposed to 2 types of commodity extraction, and 38 districts exposed to only 1 type of commodity extraction.

Extraction of construction materials occurred in 14% (*N* = 25) of all districts, metals in 12% (*N* = 22), and gemstone in 4% (*N* = 7). A total of 1% (*N* = 2) of the districts had all of the three types of commodity extraction, and 4% (*N* = 7) had at least two types of commodity extractions. A total of 45 districts (25%) were exposed to at least one type of commodity extraction. The distribution of health facilities by type of extractive activity is similar across the three types of commodity extraction (Table A2 in the Appendix A).

Table 3 summarizes the annual number of disease diagnoses per 100,000 inhabitants for different groups of diseases. Out of the main disease categories analyzed, the most common were respiratory infections and malaria, with an average of 52,704 and 33,981 cases across the five years, respectively. The least commonly diagnosed health problems were cancer and tuberculosis, with an average of 127 and 266 cases per year, respectively.

Table 4 shows the association between the type of commodity extraction and disease groups. In the fully adjusted models, we found that the presence of construction material extraction was associated with an increased incidence of chronic diseases (IRR = 1.4795% CI = 1.15–1.87). The presence of metals extraction was associated with a reduced incidence of chronic diseases (0.61, 0.47–0.80), mental health disorders (0.66, 0.47–0.92), undernutrition (0.69, 0.55–0.88), parasitic infections (0.84, 0.72–0.98), sexually transmitted diseases (0.85, 0.74–0.97), and diarrhea diseases (0.88, 0.77–0.10). No association was found between gemstone extraction and any of the main disease categories. The outputs of the crude model are available in Table A3.

Table 5 shows the results for disaggregated disease categories. Extraction of construction materials was associated with an increased incidence of hypertension (IRR = 1.49, CI: 1.11–2.01), neoplasms/cancer diseases (IRR = 1.46, CI: 1.07–1.99), bronchitis asthma (1.38, 1.14–1.66), and other cardiovascular diseases (1.72, 1.27–2.32). Gemstone extraction was associated with a significant increase in the incidence of malaria (blood slide microscopy method) (2.08, CI: 1.06–4.07), non-severe pneumonia (1.40, 1.04–1.87), diarrhea with some dehydration (IRR: 1.38, 1.11–1.73), and severe pneumonia (1.25, 1.01–1.54). Protective effects were found for metal extraction projects in the majority of disease-specific indicators.

## 4. Discussion

In this paper, we evaluated the relationship between the district-level presence of natural resource extraction activities and the disease-specific patient volumes reported across all health facilities in Tanzania in a given district and year.

Our results show that the districts’ health infrastructure across the three types of commodity extraction is relatively the same (Table A3). In addition, the presence of commodity extraction projects is associated with significant changes in the incidence of diseases reported in the routine national HMIS of Tanzania. The changes in disease incidence appear to differ substantially by the type of commodity extracted; in districts where metal resources are extracted (most typically Gold in Tanzania), we observed a significant decrease in the incidence of chronic diseases (−39%), mental health disorders (−34%), undernutrition (−31%), parasitic infections (−26%), sexually transmitted diseases (−15%), and diarrheal diseases (−13%). On the other hand, the presence of construction material extractions was associated with a significant increase in the incidence of chronic diseases (+47%), including hypertension (+49%), neoplasms/cancer (+46%), bronchitis asthma (+38%), and other cardiovascular diseases (+72%). Gemstone extraction was not associated with any of the aggregated disease categories but showed positive associations with severe and non-severe pneumonia (+40% and +25%, respectively), as well as diarrhea with dehydration (+38%).

The consistent inverse association between the metal extraction industry and disease incidence was rather striking and could be due to several factors. The observed decreased incidence of chronic diseases can partially be explained by the in-migration of young and healthy mining workers and other job seekers [47,48,49], who are at lower risk for chronic diseases, such as hypertension and diabetes mellitus [50]. In addition, the majority of large-scale metal industries in Tanzania are led by transnational corporations [51], which are more likely to adhere to international environmental and social standards (e.g., Performance Standards of the International Finance Corporation (IFC) [52]) and recommended industry practices (e.g., International Council on Mining and Metals (ICMM) [53] and Extractive Industries Transparency Initiative (EITI) [54]). Hence, it seems plausible that metal extraction projects are, on average, more able to support local health systems and community-level interventions [55] or socio-economic development programs [56]. This may also explain why, in contrast to previous studies that reported elevated levels of sexually transmitted diseases in regions where metals are extracted [15,18,57,58], our results show that districts that are exposed to metal extractions activities generally have lower incidences of sexually transmitted diseases, including HIV/AIDS, even though these differences were not statistically significant.

It is also important to highlight that our study covered the period 2015–2019, which followed two major national-level government interventions. The first intervention was the Tanzania Mineral Policy 2009 Revision, which intended to increase the mineral sector contribution to income generation [59], and the second was the re-enactment of the Mining Act 2010, which requires mining companies to have a plan to increase Tanzania’s citizens’ participation in mining activities through employment and expatriate succession plans, as well as a plan for procurement of goods and services through the available local market in the United Republic of Tanzania [60]. It is likely that these measures had a particularly positive effect on the districts with the presence of metal extractions activities, which are, in general, larger operations than construction and gemstone commodity extraction projects.

In contrast to metal extraction, increased levels of disease incidence were observed for districts with construction material or gemstone extraction activities. Extraction of construction materials covers commodities such as cement, silicate, carbonate hard rocks, decorative stones (i.e., dolomite and limestone), and industrial minerals (i.e., gypsum, clay, halite, phosphate, and mica). The observed increase in the incidence of chronic diseases appears consistent with the risk factors for such projects described by IFC, namely, high exposures to dust, vibration, noise, and unhealthy lifestyle (related to cigarette smoking and excessive alcohol intake) [61]. Studies have also shown that exposure to particulate matter (PM) from construction dust can significantly influence the appearance of diseases such as cancer and cardiovascular and respiratory diseases [62,63]. Studies from the construction and building materials literature report that there is a growing demand for construction material commodities, and they equally draw attention to the environmental and health effects this demand imposes on the livelihoods of affected communities [64,65,66]. In our study sample, 40% of the commodities extraction projects were categorized as construction materials, highlighting the need to further scrutinize this sector in terms of environmental and social responsibility, including the mitigation of potential adverse health impacts.

### Strength and Limitations

To our knowledge, this is the first large-scale study of the relationship between mining and disease incidence in Tanzania using the complete national HMIS database. The dataset covers—by definition—the entire country and allowed us to analyze a range of different mines at the same time. Having five years of data also allowed us to reduce the risk of specific results being driven by misreporting in a given month or year. The study also has several limitations. First, DHIS2 and routine HMIS data have known data quality issues [67], including concerns regarding completeness, consistency, coverage, and disease coding [68,69,70,71]. In our analysis, we did not account for potentially different data quality across districts. If such differences in quality exist, they would only bias our results if reporting was systematically correlated with the presence of mines. This is possible in principle if mines directly contribute to the health system capacity and could potentially bias results against mines. We also did not distinguish between missing data and the true zero values in our analysis, which means that the reported case numbers may underestimate the true burden of disease and patient numbers. Due to the unavailability of good data, we did not account for district per capita income, nor for the number of job seekers who migrated into mining areas. The OPD diagnoses consist of both clinical and non-clinical indicators. Non-clinical indicators can be influenced by physician’s opinion and are subject to reporting bias. As long as these biases are not correlated with the presence of mines, this should not introduce any systematic bias to our analysis. In addition, we used district boundaries to link mining operations to patient volumes. Even though districts are relatively large, it is possible that some of the mines—particularly when located near district boundaries—also affect patient volumes in neighboring districts. This exposure misclassification would likely result in less precise estimates but could also lead to an underestimation of the true causal effects if there are systematic spillovers into districts considered unexposed in our analysis. Lastly, OPD indicators represent counts of people who sought care at health facilities and do not represent actual case numbers in the population.

## 5. Conclusions

The results of this study suggest that the health impact of mining operations may not only depend on the specific health outcome but also on the types of mines. While metal extraction projects display consistent protective effects against a range of disease indicators, extraction of construction materials and gemstones is associated with increased incidence of chronic and diarrheal diseases, respectively. More research is needed to understand the underlying factors of the differences observed across the types of commodity extracted.

## Figures and Tables

**Figure 1 ijerph-18-06052-f001:**
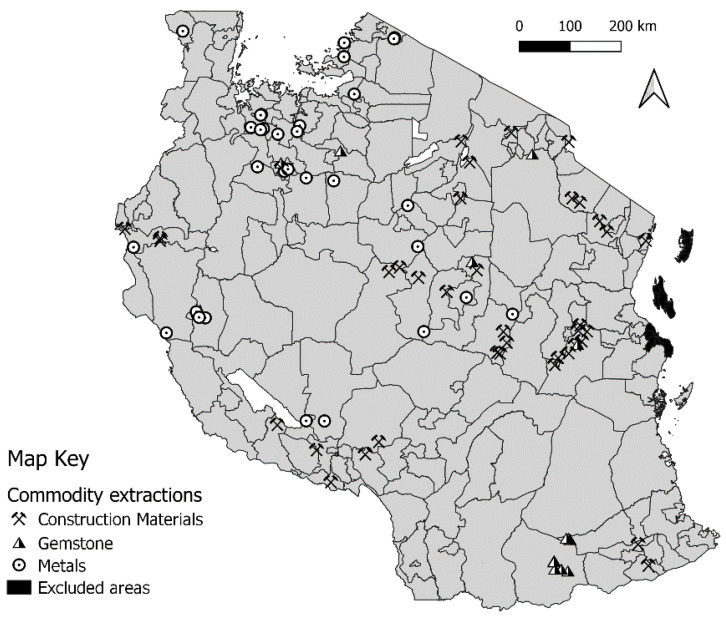
Location of selected commodity extraction projects in Tanzania.

**Table 1 ijerph-18-06052-t001:** List of disease groups and included outpatient department (OPD) indicators from DHIS2.

Disease Groups	OPD Indicator(s) Included in Disease Group
Chronic diseases	Neoplasms/cancer, hypertension, other cardiovascular diseases, diabetes mellitus, bronchial asthma
Diarrhea	Diarrhea with no dehydration, diarrhea with severe dehydration, diarrhea with some dehydration, dysentery
Sexually transmitted diseases	HIV positive ^1^, genital discharge syndrome, genital ulcer syndrome, pelvic inflammation, sexually transmitted infections
Malaria	Malaria blood smear positive, malaria mRDT positive, clinical malaria
Mental health disorders	Psychoses, neuroses
Parasitic infections	Intestinal worms, schistosomiasis
Respiratory health	Pneumonia, non-severe, pneumonia, severe upper respiratory infections
Road traffic injuries	Road traffic accidents
Tuberculosis	Tuberculosis
Undernutrition	Kwashiorkor, marasmic kwashiorkor, marasmus, moderate malnutrition

^1^ This indicator is obtained from HIV Counseling and Testing (HTC) unit. Notes: HIV stands for human immunodeficiency virus, mRDT stands for malaria rapid diagnostic test.

**Table 2 ijerph-18-06052-t002:** Overview of the number and type of health facilities and commodity extraction projects per district in Tanzania.

**Facility Type**	**Average Count**	**Median Count**	**Minimum**	**Maximum**
Hospital (*N* = 264)	1.5	1	0	11
Health center (*N* = 816)	4.6	4	1	16
Dispensaries (*N* = 6423)	36.0	35	5	83
Health clinics (*N* = 166)	0.9	0	0	23
**Type of commodity extracted**	**Number of projects**	**Number of districts with projects**	**Percent of districts with projects**	
Construction materials	38	25	14%	
Gemstone	13	7	4%	
Metals	32	22	12%	

Total number of districts = 178.

**Table 3 ijerph-18-06052-t003:** Number of disease diagnoses per 100,000 population by year.

Disease Group Indicator	Year of Observation					Average (2015/2019)
	2015	2016	2017	2018	2019	
Cancer	110	119	123	130	151	127
Cardiovascular	2643	3068	3649	4661	5840	3972
Diabetes	1020	1173	1392	1637	1970	1438
Diarrhea	11,245	11,249	11,684	12,927	13,313	12,084
HIV/AIDS (+ve test results)	1025	1231	1320	1401	1593	1314
Malaria	39,875	31,125	29,315	33,182	36,409	33,981
Mental health	434	439	500	582	662	524
Parasites	6748	6570	6727	7772	8033	7170
Respiratory infection	37,431	42,361	49,854	65,460	68,413	52,704
Road traffic accidents	811	828	785	773	795	798
Sexually transmitted infections	4720	4760	4973	6314	7410	5636
Tuberculosis	266	265	252	279	267	266
Undernutrition	476	425	342	371	350	393
Other diagnoses ^1^	51,109	53,944	58,692	71,227	83,777	63,750

^1^ Includes all other diagnoses which are not featured in any of the disease groups used for this study.

**Table 4 ijerph-18-06052-t004:** Relationship (adjusted model) between disease groups and type of commodity extraction.

Disease Groups	Type of Commodity Extraction					
	Construction Materials		Gemstone		Metals	
	IRR	CI	IRR	CI	IRR	CI
Chronic diseases	1.47 ***	1.15–1.87	1.00	0.69–1.47	0.61 ***	0.47–0.80
Respiratory infections	1.02	0.90–1.16	1.10	0.90–1.34	0.90 *	0.80–1.01
Tuberculosis	1.14	0.85–1.54	1.28	0.78–2.10	0.76 *	0.56–1.03
Diarrhea	0.97	0.84–1.12	1.25 *	0.99–1.59	0.88 **	0.77–0.99
Undernutrition	0.90	0.67–1.20	1.06	0.67–1.68	0.69 ***	0.55–0.88
Malaria	0.96	0.66–1.39	1.23	0.84–1.79	1.08	0.85–1.38
Parasitic diseases	1.08	0.93–1.26	1.10	0.83–1.47	0.84 **	0.72–0.98
Sexually transmitted diseases	1.13	0.94–1.37	1.10	0.83–1.46	0.85 **	0.74–0.97
Road traffic accidents	1.13	0.90–1.43	1.24	0.85–1.80	0.90	0.72–1.11
Mental health	1.08	0.79–1.49	1.34	0.70–2.59	0.66 **	0.47–0.92

Note: estimates are adjusted for cluster robust; *** *p* < 0.01, ** *p* < 0.05, * *p* < 0.1.

**Table 5 ijerph-18-06052-t005:** Relationship (adjusted model) between disease-specific indicators and type of commodity extraction.

Disease Groups	Type of Commodity Extraction					
	Construction Materials		Gemstone		Metals	
	IRR	CI	IRR	CI	IRR	CI
Chronic diseases						
Neoplasms/cancer	1.46 **	1.07–1.99	0.86	0.48–1.54	0.93	0.65–1.33
Hypertension	1.49 ***	1.11–2.01	1.08	0.66–1.78	0.56 ***	0.41–0.77
Other cardiovascular diseases	1.72 ***	1.27–2.32	0.74	0.43–1.27	0.62 ***	0.44–0.88
Diabetes mellitus	1.41 *	1.00–2.00	0.91	0.53–1.56	0.42 ***	0.29–0.62
Bronchitis asthma	1.38 ***	1.14–1.66	1.06	0.84–1.34	0.82 **	0.67–0.99
Respiratory infections						
Pneumonia non-severe	1.01	0.86–1.20	1.25 **	1.01–1.54	0.90	0.78–1.02
Pneumonia severe	1.04	0.87–1.24	1.40 **	1.04–1.87	0.82 **	0.67–0.99
Upper respiratory infections	1.02	0.89–1.17	1.051	0.85–1.30	0.91	0.80–1.04
Tuberculosis						
Tuberculosis	1.14	0.85–1.54	1.279	0.78–2.10	0.76 *	0.56–1.03
Diarrhea						
Diarrhea with no dehydration	0.98	0.84–1.15	1.190	0.93–1.53	0.89 *	0.77–1.02
Diarrhea with severe dehydration	0.86	0.69–1.08	1.49	0.93–2.4	0.91	0.74–1.12
Diarrhea with some dehydration	1.01	0.85–1.20	1.38 ***	1.11–1.73	0.91	0.78–1.07
Dysentery	0.85	0.61–1.18	1.46 *	0.99–2.14	0.76 *	0.57–1.01
Undernutrition						
Kwashiorkor	0.91	0.68–1.21	1.11	0.57–2.15	0.64 ***	0.48–0.86
Kwashiorkor marasmus	1.06	0.75–1.50	0.88	0.56–1.38	0.68 ***	0.52–0.90
Malnutrition moderate	0.94	0.67–1.31	0.93	0.57–1.50	0.72 **	0.56–0.93
Malaria						
Blood slide microscopy	1.11	0.70–1.73	2.08 **	1.06–4.07	0.49 ***	0.37–0.66
Clinical	1.17	0.82–1.69	1.11	0.73–1.70	0.88	0.69–1.12
MRDT	0.82	0.54–1.24	1.28	0.84–1.95	1.21	0.93–1.58
Parasite diseases						
Intestinal worms	1.08	0.92–1.27	1.06	0.79–1.44	0.82 **	0.70–0.97
Schistosomiasis	1.09	0.90–1.32	1.28	0.93–1.76	0.94	0.77–1.14
Sexually transmitted diseases						
HIV/AIDS +ve test results	1.11	0.88–1.39	1.20	0.76–1.90	0.91	0.73–1.13
Genital discharge syndrome	1.07	0.86–1.33	0.96	0.73–1.28	0.84	0.69–1.03
Genital ulcer syndrome	1.21	0.96–1.52	1.03	0.80–1.34	0.83 **	0.71–0.98
Pelvic inflammation	0.89	0.71–1.11	1.19	0.76–1.86	0.86	0.70–1.05
Sexually transmitted infections	1.12	0.94–1.34	1.36	0.93–1.99	0.81 ***	0.69–0.95
Road traffic accidents						
Road traffic accidents	1.13	0.90–1.43	1.24	0.85–1.80	0.90	0.72–1.11
Mental health						
Psychoses	1.03	0.70–1.50	1.65	0.75–3.65	0.54 ***	0.35–0.84
Neuroses	1.23	0.92–1.64	1.23	0.70–2.16	0.74 **	0.57–0.96

Note: estimates are adjusted for cluster robust; *** *p* < 0.01, ** *p* < 0.05, * *p* < 0.1; mRDT stands for malaria rapid diagnostic test, HIV stands for human immunodeficiency virus.

## Data Availability

The data presented in this study are available on request from the Ministry of Health Community Development, Gender, Elderly and Children (MoHCDGEC). The data are not publicly available due to local restrictions.

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
