# Peer review of "Associations between Natural Resource Extraction and Incidence of Acute and Chronic Health Conditions: Evidence from Tanzania"

_ijerph, 2021, doi:10.3390/ijerph18116052_

Round 1

Reviewer 1 Report

Dear authors,

the topic of your article is very interesting.

I have just two questions:

  1. if I understand well, there are a lot of incidence rates lesser than 1. This would mean that some natural resource extractions are protective for the population. Is it right?
  2. I would like to see the statistical analysis section with more details. Could you give more information about the model of Poisson? Moreover, what confounding factors have you considered in the analysis. 

Once you would have provided these details, I believe that the article could be published on the IJERPH.

Best regards.

Reviewer 2 Report

This article studies the association between the incidence of diseases and the extractive activity of natural resources and raw materials (mainly gold mining, gemstones and construction materials) in Tanzania. The design corresponds to a cross-sectional, ecological study, in which data were taken from the health information system of 178 districts in which there were extractive activities between 2015 and 2019. For association analysis, Poisson regression was used with random effects and robust cluster standard errors.

The study is well executed, the data is clear, and the article is well written. However, I have some minor comments and observations:

Please review table 2, the average count of health centers does not coincide with the text (it says 54.6 and there are 5). The lower part of the table (where the number of extractive activities according to type of material is described) is difficult to interpret without reading the text. Please specify what the parameter “% exposed” refers to, which I understand corresponds to the percentage of districts presenting each extractive activity. Actually, table 2 could be separated in two tables.

Also, it would be interesting to make another table that shows how the different types of health facilities are distributed according to the type of extractive activity in the districts (only metals, only construction materials, only precious stones, or the combination of these types). This would help to interpret the data or to answer the question if lower incidence of diseases in the districts where there is metal extraction is due to better health infrastructure. It would also be interesting to consider the per capita income in the districts studied to enrich the discussion. What does the immigration of healthy workers refer to in the districts where there is metal mining? Perhaps this could be better explained.

The text description of table 3 does not agree with the title of the table. The text says that “Table 3 summarizes the average number of patients registered at the district level in each year”, when actually it shows the annual incidence per 100,000 inhabitants of different groups of diseases.

Other minor comments:

Page 2, last paragraph, line 90, the word “district” is repeated.

Page 3: last paragraph, line 119, there should be a period instead of a dash.

Page 8, third paragraph, line 230, there should be a comma instead of a period after the quote [59].
